# Five Functional Aspects of the Epidermal Barrier

**DOI:** 10.3390/ijms222111676

**Published:** 2021-10-28

**Authors:** Alain Lefèvre-Utile, Camille Braun, Marek Haftek, François Aubin

**Affiliations:** 1Sce de Pédiatrie Générale et Urgence pédiatrique, Hôpital Jean Verdier, Assistance Publique Hôpitaux de Paris, 93140 Bondy, France; alain.lefevreutile@gmail.com; 2Unité 976 HIPI, Institut de Recherche Saint-Louis, Université de Paris, Inserm, 75010, Paris, France; 3Centre international de Recherche en Infectiologie, Inserm U1111, Université Claude Bernard Lyon 1, 69007 Lyon, France; camille.braun@inserm.fr; 4Sce de Pneumologie Pédiatrique et Allergie, Hôpital Femme Mère Enfant, 69500 Bron, France; 5CNRS UMR5305, Laboratory of Tissue Biology and Therapeutic Engineering, LBTI, Lyon1 University, 69100 Lyon, France; marek.haftek@univ-lyon1.fr; 6Inserm U1098, Université de Franche Comté, 25000 Besançon, France; 7Sce de Dermatologie, Centre Hospitalier Universitaire, 25000 Besançon, France

**Keywords:** epidermis, skin barrier, stratum corneum, ceramides, keratinization, desmosomes, tight junctions, immunity, sensory nerves, microbiome

## Abstract

The epidermis is a living, multilayered barrier with five functional levels, including a physical, a chemical, a microbial, a neuronal, and an immune level. Altogether, this complex organ contributes to protect the host from external aggression and to preserve its integrity. In this review, we focused on the different functional aspects.

The skin, measuring around 2 m^2^ in an adult, is our largest organ and provides our organism with integrity and identity. It also allows exchange with our environment, while simultaneously mediating protection from it. The most important function of the skin is to provide an effective barrier between the internal and external environments of an organism [1,2,3,4,5,6,7]. The epidermis provides an interface/outside-inside barrier that contributes to three primary functions: limiting passive water loss, protecting against environmental aggressions (e.g., chemical, ultraviolet light, allergens…), and preventing microbial infection. In addition, the epidermis should be able to mount an effective regenerative response after injury, since integrity of the skin barrier is required to fulfill its functions [8]. Five functional aspects of the cutaneous barrier can therefore be distinguished, including a physical, a chemical, a microbial, a neuronal, and an immune level.

## 1. Physical Protection

The epidermis is characterized by a highly cohesive, multilayered cell architecture. Living cell layers of the epidermis are crucial in the formation and maintenance of the barrier on two different levels. First, keratinocytes form the outermost protective, dead layer of the skin—the cornified or horny layer (*stratum corneum*, SC), through a complex terminal differentiation process named keratinization. Second, the living cell layers form a resilient tissue by providing mechanical cohesion between cells by means of desmosomes, adherens junctions (AJ), and tight junctions (TJ) (Figure 1).

### 1.1. Keratinization

The epidermis is a stratified epithelium composed of morphologically distinct cellular layers that reflect the keratinocyte terminal differentiation process. Depending on the anatomical location, these layers from deep to superficial are: *stratum basale*, *stratum spinosum*, *stratum granulosum*, *stratum lucidum*, and *stratum corneum* (SC).

The *stratum lucidum* is morphologically distinguishable only in the thick and hairless skin of palms and soles. It is composed by two to three layers of dead and anucleate keratinocytes (corneocytes) presenting particular colorless staining properties at the basal part of the horny layer. Within the epidermis, keratinocyte proliferation is restricted to the basal cell layers. After mitosis, keratinocytes progressively differentiate and migrate through the epidermis towards skin surface, to finally lose their nuclei and other cellular organelles, cornify and flatten, thus forming the SC [2]. The process of keratinization is accompanied by expression of different constitutive proteins characteristic for each differentiation stage (Figure 1).

Keratin (K) is the main structural protein, contributing to 30%–80% of the total epidermal proteins. More than 50 mammalian keratins have been identified and characterized. Keratins can be sub-classified into two distinct classes: Type I keratins, including K9 to K40, are relatively acidic whereas type II keratins, including K1 to K8 and K71 to 86, are more basic. The active keratin genes are clustered into two dense regions of the chromosomes 12q and 17q [9]. Keratins are arranged by pairs belonging to the two aforementioned families to form intermediate filament bundles of the cytoskeleton.

In the *stratum granulosum*, keratinocytes contain filaggrin (FLG)-rich keratohyalin granules (KHGs) with filaggrin (FLG) and lamellar granules (LGs, lamellar bodies) containing lipids, proteases, protease inhibitors, antimicrobial peptides (AMP), and some structural proteins (e.g., corneodesmosin). Granular keratinocytes are crucial to SC homeostasis and barrier function through (1) the production and release of LG content to the extracellular spaces beneath SC [10,11] and (2) the liquid-liquid phase changes and biochemical transformations of KHG proteins [4].

As the outermost layer of the skin, the SC is the first line of the outside-in cutaneous barrier. The SC is composed of flattened, multilayered anucleate keratinocytes called corneocytes, surrounded by multiple planar lamellae sheets of lipid lamellae, enriched in ceramides, cholesterol, and free fatty acids (FFA) [12,13]. Cornification is the last stage of terminal differentiation leading to cell death. All metabolic activity ceases, cells lose their organelles including the nucleus, and lipids and proteins are extruded to fill the spaces between cornified cells, effectively sealing the skin. At the skin surface, dead corneocytes are progressively shed off and the relatively constant thickness of the SC is maintained through continuous keratinization at the bottom [14].

Although evidence exists for a role of organelle disintegration, proteases, nucleases, and transglutaminases contributing to cell death, the molecular coordination of this mode of programmed cell death is far from understood [15,16,17]. Disorganized apoptotic or necrotic cell death would lead to the release of DAMPs (danger-associated molecular patterns including self antigens), inflammation autoimmunity, and disturbance of the differentiation process of adjacent keratinocytes. Interestingly, it has been demonstrated that keratinocytes may activate non-apoptotic and non-necroptotic pathways, i.e., autophagy, to prevent premature cell death during terminal differentiation. In addition, autophagy is involved in the immune response and neutralizes pathogens and clears senescent cells [15,16,17]. However, the autophagy machinery can either protect or cause autoimmune disorders. The physiological circumstances that govern this “balancing act” are not well understood. The cytokine microenvironment from dying immune cells, pathogens, or senescent cells can potentially direct the autophagic response in skin. It has been suggested that autophagy may participate in various physiological activities to ensure the smooth and quiescent operation of the immunotolerant environment and to maintain skin integrity [16].

The “brick and mortar model” has been proposed to illustrate the SC architecture [14,18]. Protein and lipid envelopes surround each brick, i.e., the corneocyte. The corneocyte protein envelope includes mainly loricrin, involucrin, small proline-rich proteins (SPRPs), and filaggrin (FLG), which are substrates for the enzymatic action of transglutaminases [19,20]. The corneocyte lipid envelope is a monolayer of ceramides that serves as a scaffold for the organization of intercellular lamellar lipids of the SC. The “mortar” component is a lipid-rich matrix composed of ceramides, cholesterol, and FFA modified by different SC enzymes. The SC extracellular matrix contains not only lipids but also enzymes, structural proteins, and antimicrobial peptides [21]. The inter-corneocyte lipids/proteins matrix and protein-rich corneocytes are critical for the formation of the functional skin barrier.

A major component of the barrier is the multifunctional epidermal protein FLG. FLG derives from a large precursor, pro-FLG contained into the KHGs of the *stratum granulosum* keratinocytes [22]. As keratinocytes cornify, pro-FLG undergoes dephosphorylation and enzymatic processing to deiminated filaggrin molecules dispersed in the corneocyte interior and becomes partially crosslinked to cornified envelopes [23,24]. Ultimately, free amino acids resulting from the filaggrin breakdown form the major part of the so-called natural moisturizing factor (NMF) of SC, presenting an exceptional water-holding capacity and contributing to the low pH of SC [25]. Recently, physico-chemical changes in liquid-liquid phase transition accompanying acidification of the cell contents at the interface between living epidermis and SC, have been incriminated in the dispersion of proteins condensed in KHGs upon cornification [24].

Ceramide is a general term for certain well-characterized sphingolipid metabolites and second messengers involved in numerous biological processes in the cell. Ceramides are not only a key molecules in sphingolipid metabolism but are also important signaling molecules that are able to regulate vital cellular functions [26]. They consist of a backbone (sphingosine (S), phytosphingosine (P), or 6-hydroxy-sphingosine (H)) and a fatty acid residue (non-hydroxy fatty acid (N), hydroxy fatty acid (A), or esterified hydroxy fatty acid (EO)). The combination of a fatty acid and a backbone results in various subclasses of ceramide fractions found in the SC [26,27]. SC ceramides are an essential element of the extracellular lipid matrix and play a critical role in the formation of the most important natural barrier to transepidermal water loss and to penetration of various compounds into the skin. Intercellular lipid multilayers, composed of quasi-equimolar proportions of ceramides, free fatty acids, and cholesterol, form hydrophobic stacks between SC corneocytes. Yet, hydrophilic inclusions are present within this crystalline-like extracellular matrix. They carry proteins delivered to the extracellular spaces at the same time as lipids and thus may swell upon SC hydration [28]. In this way, a humid environment promotes desquamation by facilitating accessibility of junction-degrading enzymes and promotes transepidermal fluxes of water soluble substances. Inflammatory skin diseases are accompanied by a shift in the proportion of ceramide subclasses. In the SC of atopic dermatitis (AD) patients, the ceramide subclass AP is increased while in psoriasis lesions, the subclasses AP, NP, and EOP are lowered [26], witnessing to the abnormal inside-out and outside-in SC barrier.

### 1.2. Epidermal Cell Junctions

To obtain a fully functional skin barrier, keratinocytes should be connected to each other by intercellular junctions that link intercellular contacts to the cytoskeleton (Figure 1) [29].

Adherens junctions (AJs) are intercellular structures that couple cell membrane to the cytoskeleton of actin, thereby creating a transcellular network. The molecular basis of AJs consists of two cell adhesion receptor complexes, the classical cadherin/catenin complex and the nectin/afadin complex, which both link to the actin filaments of the cytoskeleton. Two types of classical cadherins are expressed in the epidermis: P-cadherin (cadherin 3) in the basal layer, and E-cadherin (cadherin 1) in all living layers of the epidermis. In vitro and in vivo studies using knockout/knockdown strategies demonstrated that cadherins are also involved in the formation of TJs and desmosomes [30,31,32].

Tight Junctions (TJs) are occluding junctions composed of three major transmembrane proteins, i.e., occludin, claudins, and E-cadherin [33]. These associate with different peripheral cell proteins such as zonula occludens (ZO) located on the intracellular side of the membrane, which anchor to the actin filaments of the cytoskeleton [34]. TJs constitute the secondary outside-in barrier, next to the SC, and are primarily involved in the regulation of inside-out water fluxes.

Gap Junctions (GJs) establish a specialized communication system within the epidermis suitable for the exchange of ions, metabolites, and secondary messengers. GJs are aggregates of protein channels formed by adjacent cells that allow direct communication between the cytosols. These intercellular channels assemble from various connexin subunits named after the molecular mass of each family member. Six connexins oligomerize within a cell to form a hemichannel (connexon) that is transported to the plasma membrane and connects with a similar structure on the other cell. Connexins are involved in keratinocyte proliferation and are upregulated in wound healing, psoriasis, and skin cancer [35,36]. Electrical and metabolic coupling provided by GJ is important for coordination of the differentiation processes and, in fine, constitution of the SC barrier.

Desmosomes are composed of the transmembrane desmosomal cadherins, desmogleins (DSG) 1–4 s, and desmocollins (DSC) 1–3 [37] (Figure 2). The intracellular ends of desmosomal cadherins are inserted in the molecular network of adaptor proteins forming desmosomal plaques, to which keratin filaments bind. Two families of plaque proteins can be distinguished in desmosomes: (i) plakoglobin and plakophilins and (ii) desmoplakin, envoplakin, periplakin, and plectin from the plakin family. According to the level of keratinocyte differentiation, lower epidermal DSG 2 and 3 are progressively substituted by DSG 1 and 4 in the upper viable epidermal layers. In the same way, DSC 3 is replaced by DSC 1 (Figure 1). In the SC, the morphology of desmosomes changes dramatically, and they are then called corneodesmosomes [38].

The layered structure of the intercellular portion of the junction is lost, whereas the intracellular plaque becomes embedded within the cross-linked cornified envelope. DSG 1, DSC 1, and corneodesmosin (CDSN) are structural components of the corneodesmosome cores. Desquamation, i.e., the release of corneocytes at the skin surface occurs after cleavage of DSG1, DSC1, and CDSN by various lamellar granules-secreted proteases, including kallikreins (KLKs) and cathepsins. This process is regulated by protease inhibitors expressed in the SG and also transported to the extracellular space of SC by lamellar granules [39,40].

Accelerated desquamation and degradation of corneodesmosomes caused by either overexpression of proteases or downregulation of their inhibitors result in epidermal barrier dysfunction observed in different diseases [41].

Genetic mutations provoking impairment of keratinocyte differentiation or targeting epidermal junctions (Table 1 and Table 2) are associated with different skin phenotypes [40,42,43,44]. Obviously, defective SC formation and/or its precocious desquamation lead to severe abnormalities in epidermal barrier function.

### 1.3. Protection against Ultraviolet Radiations (UVR)

Ultraviolet radiation penetrates skin to variable depths, depending on the wavelength: long-wave UVA is responsible for dermal actinic ageing whereas short-wave, highly energetic UVB rays can reach no further than living cell layers of the epidermis but cause acute (sunburn) and long-term damages (cumulative changes to the genetic material). SC is able to absorb a sizeable part of UVB energy, thanks mainly to the presence of various chromophores, including trans-urocanic acid—one of the products of FLG degradation [45].

Melanin, the pigment elaborated by melanocytes, is another adaptive filter to UVR. Melanocytes are found between keratinocytes of the stratum basale, at the ratio of 1:10, and form the *epidermal melanin* units as a result of the relationship between one melanocyte and 30–40 associated keratinocytes. Melanin is synthesized in melanosomes—specialized, lysosome-like organelles. Mature melanosomes are transported along melanocyte microtubules within the pigment cell dendrites and transferred to neighboring keratinocytes [46]. These melanosomes form a cap over the cell nucleus to protect keratinocytes’ genetic material from being damaged by UVR (Figure 3) [47]. Epidermal melanin is a primary absorber of UVR, thus protecting the underlying epidermal and dermal elements. Melanin absorbs both ultraviolet and visible light and transforms the energy into heat through internal conversion. The absorption increases linearly in the range of 720 to 620 nm and then exponentially toward shorter wavelengths (300–600 nm). Upon UVB exposure, delayed pigment darkening appears in the skin. The tanning response is probably the most striking photo-protective mechanism against the detrimental effects of UVR exposure to the skin. Stimulation of the UV-dependent α-MSH-MC1R pathway activates the cAMP-CREB-MITF cascade in melanocytes, resulting in the synthesis of melanin and eventual transfer of melanosomes to keratinocytes for protection against UVR [48,49]. Epidermal (and follicular) melanocytes produce two types of melanin pigment, eumelanin and pheomelanin. Epidermal photoprotective barrier to UVR is based on the predominant pigment, eumelanin, which functions as a free radical scavenger with superoxide dismutase-like activity that reduces reactive oxygen species (ROS). Pheomelanin, instead, can generate ROS through UV-dependent and UV-independent pathways, exposing therefore fair-skinned individuals to an increased risk of melanoma [50].

## 2. Chemical Aspect of the Epidermal Barrier

The chemical factors included in the SC contribute to its acidic pH and to the antimicrobial activity through the release of anti-microbial peptides (AMP).

The pH of normal human SC ranges from 4.5 to 5.5 and is involved in the permeability barrier homeostasis, the cohesion of the SC, the regulation of the microbiome, and the down-regulation of pro-inflammatory cytokine signaling [51]. Different endogenous mechanisms account for the global reduction in the pH of the SC, including the hydrolyzation of phospholipids into FFA by phospholipases (sPLA2), the acidification by the sodium-hydrogen antiporter type 1 (NHE1), the catabolism of FLG into free amino acids including trans-urocanic acid and the release of protons from melanin granules.

The natural-moisturizing factors (NMFs) of SC is produced upon cornification. It comprises amino acids and their derivatives (pyrrolidone carboxylic acid and urocanic acid) resulting from the proteolysis of epidermal FLG. Changes in the NMFs alter the epidermis pH and lipids, indicating an interdependence between the chemical and the physical barrier functions [52,53,54]. Other components of the NMFs include lactates, urea, and electrolytes which also contribute to the pH and the state of hydration of the SC. Secretions from the sebaceous glands, containing triglycerides, wax esters, and squalene, are delivered directly onto the top of SC and add up to the water holding capacity and maintenance of the low pH. Indeed, bacteria and yeasts from the epidermis microbiome hydrolyze triglycerides into FFA, contributing to the acidification of the skin.

Sweat is not only involved in thermoregulation but is also able to increase and maintain skin hydration [55]. Urea and lactic acid present in sweat help in moisture retention by SC and make part of NMF. Eccrine sweat contains a large number of minerals, proteins, proteolytic enzymes, AMPs, and various inflammatory cytokines, such as IL-1, IL-6, and IL-31, which can act as danger signals by activating keratinocytes [56]. The acidic mantle at the skin surface is also partially constituted by sweat. Kazal-type 5 (SPINK5) protease inhibitor, which contributes to maintaining epidermal homeostasis, and cysteine A protease inhibitor, which serves as the first line of defense against allergens with cysteine protease activity, are also secreted with sweat. As sweat is delivered onto the skin surface, it has no direct contact with living keratinocytes in healthy skin. However, in the skin with a defective cutaneous barrier, the leakage of sweat into the epidermis and the dermis causes not only chronic inflammation associated with itching sensation but also dry skin [57].

## 3. Role of the Microbiome in Skin

The skin’s natural microbiome colonizes the epidermal surface (Figure 4). The composition of the cutaneous microbiome includes bacteria, fungi, and viruses, and is fairly stable. Culture-independent genomic approaches have shown that, in contrast to the gut microbiome, the skin microbiota are dominated by *Actinobacteria* with an abundance of Gram-positive bacteria, such as *Staphylococcus,*
*Propionibacterium*, and *Corynebacterium* species [58]. The composition of microbial communities depends on the skin site, with changes in the relative abundance of bacterial taxa associated with moist, dry, and sebum-rich microenvironments. Sebaceous sites are dominated by lipophilic *Propionibacterium* species, whereas bacteria that thrive in humid environments, such as *Staphylococcus* and *Corynebacterium* species, are preferentially abundant in moist areas, including the bends of the elbows and the feet.

Bacterial commensal colonization of human skin is vital for the defence-training and maintenance of the skin’s innate and adaptive immune functions [59]. *Staphylococcus epidermidis*, belonging to the natural cutaneous microbiome, inhibits colonization by the pathogenic *Staphylococcus aureus* and induces expression of AMPs resulting in *S. epidermidis*-orchestrated innate immune alertness. Additionally, it may enhance epidermal barrier function through the increased expression of TJ proteins occludin and ZO-1 [60].

Tight regulation of the composition of the microbiome is provided by the epidermis pH and by the skin immune system. Efficacy of the epidermal barrier is, thus, the outcome of the threefold crosstalk between the chemical barrier, the skin immune system, and skin microbiota. The skin microbiome participates in transmission of xenobiotic environmental signals to the functional immune network of the skin [61].

## 4. Immunological Aspects of the Barrier Function

The immune barrier of the skin comprises a variety of resident immune cells populating epidermis and dermis (Figure 4) [1]. Constitutive sentinels of the immune system, such as several types of resident antigen-presenting cells e.g., *epidermal* Langerhans cells and dermal dendritic cells, cooperate with keratinocytes and adaptive tissue-resident memory lymphocytes to maintain the surveillance against foreign aggressions [62].

An important part of this barrier belongs to the innate defense system. AMPs are considered a rapid and first-line response of the innate immune system to microbial pathogens. Together with their antimicrobial effects, AMPs also exert immunomodulatory effects by inducing cell migration, proliferation, and differentiation, regulating cytokine/chemokine production, and sustaining the barrier function of the skin [63]. AMPs, such as LL-37, human beta-defensins, and S100A7 are produced and released by keratinocytes and immune cells [64]. This non-specific immunity is effective against a wide range of pathogens and contributes to the regulation of the skin microflora [65].

Altogether, this immune armada efficiently senses microbial danger signals via pathogen- and damage-associated molecular patterns (PAMPs and DAMPs) and specific pattern recognition receptors that include Toll-like receptor (TLR), Nod-like receptor (NLR), and C-type Lectin receptor (CLR). Initiation of an adequate immune response results in tissue inflammation and further barrier disruption aimed at clearing the xenobiotic invasion. Beyond this necessary but locally harmful action, resident immune cells subsequently contribute to skin barrier repair and re-establishment of homeostasis.

Several feedback mechanisms exist between various functional components of the skin barrier. The immune defense system is influenced by other skin barrier components and vice versa. AMP expression is thus regulated by the cutaneous microbiome. Activation of various TLRs enhances TJ barrier function by up-regulation of the expression of claudins, ZO-1, and occludin [66]. In contrast, proinflammatory cytokines like interleukin 1β and tumor necrosis factor alpha have been shown to differently modulate TJ function [67]. Furthermore, TJs together with the functional SC barrier are involved in the regulation of innate immunological barrier because they limit the contact of living epidermal cells expressing pattern recognition receptors with the external danger signals including pathogens [68].

## 5. Participation of the Sensory Neuronal System in the Barrier Function

The skin is a sensory organ that comprises a large range of sensory receptors associated with non-neuronal components (Figure 4), such as Meissner corpuscles, Pacini corpuscles, and Ruffini endings in the dermis or Merkel cell complexes in the epidermis [69,70]. Besides, intra-epidermal nerve fibers (IENF), classified as C- and Aδ-fibers, unmyelinated, and thickly myelinated, respectively transduce stimuli via specific receptors, in particular Transient Receptor Potential (TRP) ion channels [71]. Originally described as “touch corpuscles” by Friedrich Merkel in 1875, Merkel complexes are groups of specialized cells in the epidermis of glabrous skin that are innervated by sensory fibers, via synaptic junctions. Merkel cells are anchored within the epidermis by thin cytoplasmic protrusions projecting to keratinocytes and by desmosomes [70]. The IENF are not the exclusive transducers of pain and itch. While TRP is expressed both by sensory neurons and keratinocytes, its specific and selective activation on keratinocytes is sufficient to induce pain [71]. The entire epidermis may be thus considered as a sensory tissue [72]. IENF can communicate with different cell populations in the different layers of the skin by releasing various types of neuropeptides. Almost all cutaneous cells express functional receptors for neuropeptides, through which they receive signals from the nervous system. In turn, skin cells produce neuropeptides and neurotrophins, which stimulate nerve fibers. This exchange creates a positive bidirectional feedback loop. There is increasing evidence that the cutaneous nervous system modulates physiological and pathophysiological effects including cell growth and differentiation, immunity, and inflammation as well as tissue repair. Skin innervation is part of the peripheral nervous system. This cutaneous nervous system is constantly receiving and responding to various types of stimuli which can be either physical (thermal, mechanical, electrical, light), chemical, or produced by allergens, haptens, microbiological agents, trauma, or inflammation. Cutaneous nerves can also respond to stimuli from the blood stream and react to emotions. Moreover, the central nervous system can modulate a large number of functions within the skin including vasomotricity, thermoregulation, piloerection, gland and cell secretion, tissue growth and differentiation, wound healing, immune response, and inflammation [73,74]. This happens either directly, via efferent autonomic nerves and brain-derived mediators or indirectly, through the immune cells and adrenal glands relaying central signals.

The epidermis is thought to protect sensory nerves from overexposure to environmental stimuli, and epidermal barrier impairment leads to pathological conditions associated with itch, such as atopic dermatitis. Epidermal nerve endings are pruned through interactions with keratinocytes to stay below the TJ barrier, and disruption of this mechanism may lead to aberrant activation of epidermal nerves and pathological itch [75].

The epidermis closely interacts with nerve endings and both epidermis and nerves produce substances for mutual sustenance. Several factors including pH gradient, skin barrier integrity, irritant exposure, and the microbiome can modulate these interactions. Complex interplay exists between the immune and sensory nervous systems through the activation of protease-activated receptors (PARs) by endogenous and microbiome-derived proteases and the release of cytokines, neuropeptides, and neurotrophins by both keratinocytes and neurons [73,76]. Activation of PARs can be linked to downstream activation of transient receptor potential (TRP) of ion channels, e.g., ENaC that mediate neurogenic inflammation and pain and participates to the production and the secretion of lipids by the epidermis [77]. Furthermore, PAR2 might contribute to epidermal barrier impairment by compromising TJs integrity and claudin-1 expression [78].

## 6. Conclusions

The epidermis provides a protection against mechanical, chemical, and microbial injury through the formation of various interactive functional barriers.

If one barrier system is altered and not compensated, this may lead to a vicious circle of inflammation and the development of skin diseases such as AD [52,54]. A major challenge for future treatment approaches will be to understand the interdependence of these five functional parts of the cutaneous barrier and to apply specific regimens that restore its proper function [1,3]. It has been recently demonstrated that the phosphodiesterase-4B inhibitor crisaborole, the IL4 receptor antagonist dupilumab, and JAK-STAT inhibitors which are involved in the regulation of the immunological barrier are also able to increase the expression of filaggrin, loricrin, and claudins, thus contributing to the restoration of the physical barrier [79,80,81].

## Figures and Tables

**Figure 1 ijms-22-11676-f001:**
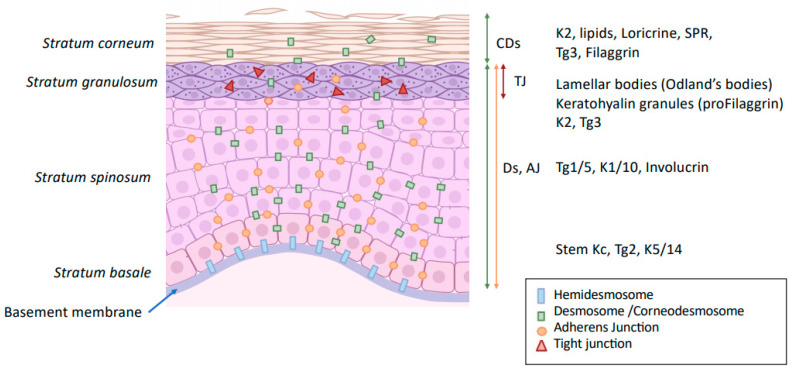
Modification of intercellular junction along keratinocyte differentiation. Kc: keratinocytes Ds: desmosome; CDs: corneodesmosome; TJ: tight junction; AJ: adherens junction; K: keratin; Tg: transglutaminase; SPR: small proline-rich proteins.

**Figure 2 ijms-22-11676-f002:**
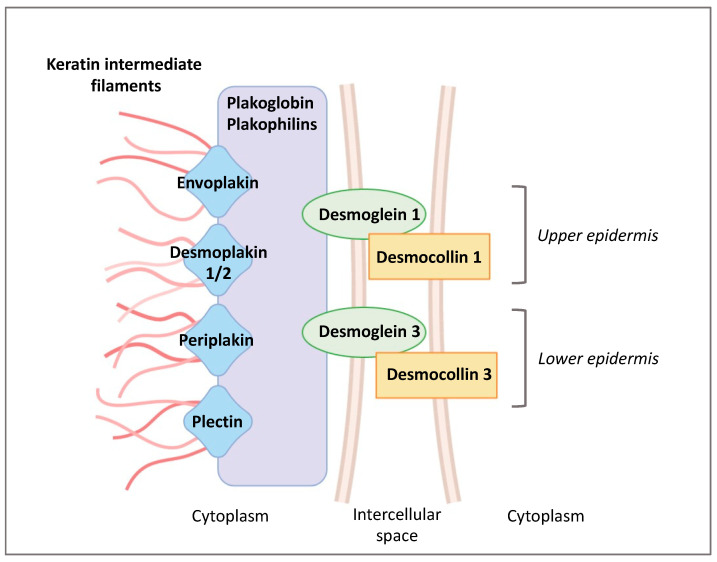
Changes in desmosome composition during keratinocyte differentiation.

**Figure 3 ijms-22-11676-f003:**
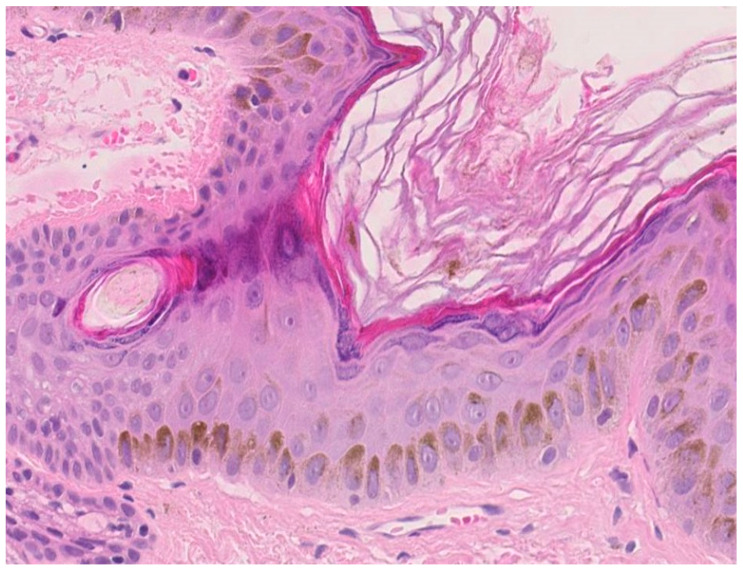
Human epidermis after solar irradiation (magnification × 120, HES stain). UVB light stimulates melanin secretion which acts as a built-in sunscreen. Epidermal melanin is a primary absorber of ultraviolet radiation, thus protecting the underlying epidermal and dermal elements.

**Figure 4 ijms-22-11676-f004:**
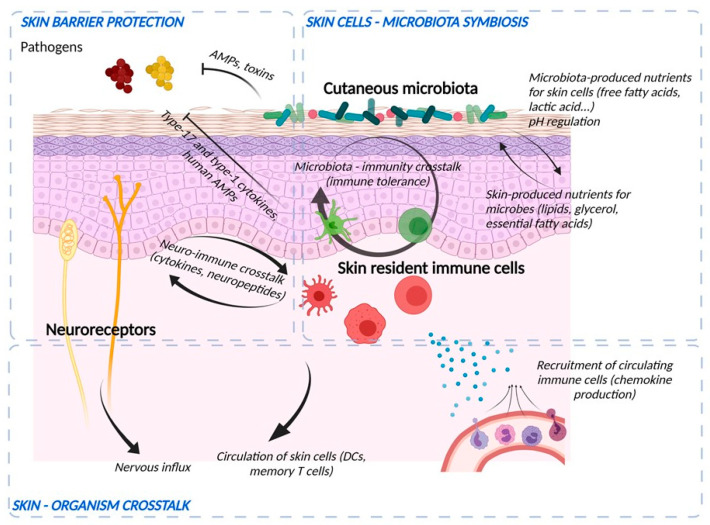
Crosstalk between microbiological, immunological, and sensory nerve barriers of the epidermis. Bacteria and skin cells are organized in a symbiosis in which nutrients are shared and protection from environmental hazards (e.g., pathogens) is provided by the different actors. Bacteria produce their own AMPs and stimulate immune cells and keratinocytes to induce immune vigilance and modulate the epidermal structure. In return, the epidermis produces nutrients and ligands essential for the establishment of commensal flora and develops an anti-pathogen defense via the production of AMPs, the action of cells such as neutrophils and macrophages, the production of cytokines to potentiate the action of these cells and chemokines to recruit them from the bloodstream. Immune cells and keratinocytes also interact with nerve endings (“neuroinflammation”) to potentiate the response to pathogens and participate in sensory awareness. Finally, the skin barrier informs the body by sending nerve impulses and circulating skin cells to lymphoid organs.

**Table 1 ijms-22-11676-t001:** **Epidermal Differentiation-related Genetic Disorders.** Adapted from Oji et al. [42]. Syd: syndrome; K: keratin; AR: autosomal recessive.

Disease	Localization	Molecule/Function	Gene
Epidermolysis bullosa simplex	**Keratins**	K5/K14	*KRT5/KRT14*
Epidermolytic ichthyosis of Brocq		K1/K10	*KRT1/KRT10*
Epidermolytic ichthyosis of Siemens		K2	*KRT2*
Curt-Macklin syd		K1	*KRT1*
Ichthyosis variegata		K10	*KRT10*
Epidermolytic keratoderma (Vörner-Thost)		K9	*KRT9*
White sponge hyperplasia		K4/K13	*KRT4/KRT13*
Ichthyosis vulgaris	**Keratohyalin granules**	Filaggrin	*FLG*
CEDNIK syd	**Lamellar granules**	Fusion of membranes	*SNAP29*
Harlequin ichthyosis		Transporter	*ABCA12*
Conradi-Hunermann-Happle syd		Cholesterol synthesis	*EBP*
X-linked ichthyosis		Steroid sulphatase	*STS*
Chanarin Dorfman syd		Lipid metabolism	*ABHD5*
Netherton syd		LEKTI (protease inhibitor)	*SPINK5*
Sjögren-Larsson syd		Lipid metabolism	*ALDH3A2*
Papillon Lefevre syd		Cathepsin C	*CTSC*
AR congenital ichthyosis	**Cornified envelope**	Transglutaminase	*TGM1*
Progressive symmetric erythrokeratoderma		Loricrin	*LOR*
Vohvinkel syd with ichthyosis		Loricrin	*LOR*

**Table 2 ijms-22-11676-t002:** **Epidermal Junctions-related Genetic Disorders****.** Adapted from Haftek [40], Guerra et al. [43] and Petrof et al. [44]. PPK: Palmo-Plantar Keratoderma; ARVC: Arrhythmogenic right ventricular cardiomyopathy; syd: syndrome.

Disease	Junction	Molecule	Gene
Peeling skin syd type B	**Corneodesmosome**	Corneodesmosin	*CDSN/PSOR1*
Hypotrichosis simplex			*CDSN*
Skin dermatitis, multiple severe allergies, metabolic wasting syd Type I striate PPK	**Desmosome**	Desmoglein 1	*DSG1* *DSG1*
Localized Hypotrichosis		Desmoglein 4	*DSG4*
ARVC cardiomyopathy with PPK and woolly hair		Desmocollin	*DSC2*
DSC3 and hypotrichosis and recurrent skin vesicles			*DSC3*
Type II striate PPK ARVC 8 Carvajal syd: striated PPK, woolly hair, and left ventricular cardiomyopathy Skin fragility/woolly hair syd Lethal acantholytic epidermolysis bullosa		Desmoplakin	*DSP*
ARVC 12 Naxos disease: ARVC, PPK, woolly hair ARVC, PPK, alopecia Focal and diffuse PPK, woolly hair Lethal congenital epidermolysis bullosa		Plakoglobin	*JUP*
Ectodermal dysplasia-skin fragility syd		Plakophilin	*PKP 1*
Hailey Hailey’s disease		Ca(2+)/Mn(2+)-ATPase (SPCA1)	ATP2C1
Darier’s disease		Calcium/ATPase (SERCA2)	*ATP2A2*
NISCH syd	**Tight Junctions**	Claudin 1	*CLDN1*
KID syd Keratoderma hereditaria mutilans (Vohwinkel’s syd) Hidrotic ectodermal dysplasia (Clouston syd) Erythrokeratoderma variabilis (Mendes da Costa syd)	**Gap Junctions**	Connexin 26 Connexin 26 Connexin 30 Connexin 31	*GJB2* *GJB2* *GJB6* *GJB3*

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
