# Peer review of "Five Functional Aspects of the Epidermal Barrier"

_ijms, 2021, doi:10.3390/ijms222111676_

Round 1

Reviewer 1 Report

This is a nice review about the skin’s barrier and homeostasis function. Five barriers are distinguished: physical barrier function, chemical barrier function, microbial barrier function, immunological barrier function and sensory nerve barrier function. The review is nicely structured and covers all important points of the biology of the skin as barrier.

Some minor comments:

There are some typos in the text

Citations have to be reviewed. (e.g. there should be a reference at least after each paragraph; reference in subheading? Page 3 line 113)

The abbreviations on page 3 in line 112 are not introduced.

The figure legends should be more detailed.

Please review the cross-references for the figures.

Please review parenthesis (are sometimes not closed).

Please review the author contributions – the initials do not fit.

Page 11 line 353: … the IL-4 receptor alpha agonist Dupilumab …

Please review the structure of the manuscript: number ALL or NONE of the barrier headings

Author Response

Response to reviewer #1

General comment: This is a nice review about the skin’s barrier and homeostasis function. Five barriers are distinguished: physical barrier function, chemical barrier function, microbial barrier function, immunological barrier function and sensory nerve barrier function. The review is nicely structured and covers all important points of the biology of the skin as barrier.

We thank reviewer #1 for these encouraging feedbacks and hope that the revised version of the manuscript will support his opinion

Some minor comments:

Comment 1: There are some typos in the text

We thank the reviewer for his/her careful reading. Accordingly, we extensively read the revised version of the manuscript to remove all typos we could detect.

Comment 2: Citations have to be reviewed. (e.g. there should be a reference at least after each paragraph; reference in subheading? Page 3 line 113) and Comment 3: The abbreviations on page 3 in line 112 are not introduced.

As suggested, corrections were made on references and abbreviations. In addition, we added the following information to explain the abbreviations: “They consist of a backbone (sphingosine [S], phytosphingosine [P], or 6-hydroxy-sphingosine [H]) and a fatty acid residue (non-hydroxy fatty acid [N], hydroxy fatty acid [A], or esterified hydroxy fatty acid [EO]). The combination of fatty acid and backbone results in 12 subclasses of ceramide fractions of the stratum corneum”. Please refer to lines 116 to 119 of the revised manuscript.

Comment 4: The figure legends should be more detailed.

We thank the reviewer for this comment. Figure legends have been expanded for better clarity and information.

Comment 5: Please review the cross-references for the figures.

We thank the reviewer to point this out. Figure references have been reviewed and corrected when needed.

Comment 6: Please review parenthesis (are sometimes not closed).

We thank the reviewer to point this out. Parentheses have been reviewed and corrected when needed.

Comment 7: Please review the author contributions – the initials do not fit.

We thank the reviewer for this comment. Initials have been corrected.

Comment 8: Page 11 line 353: … the IL-4 receptor alpha agonist Dupilumab …

We thank the reviewer for this comment. We added the following information for dupilumab: « the IL4 receptor antagonist ». Please refer to line 347 of the revised manuscript.

Comment 9: Please review the structure of the manuscript: number ALL or NONE of the barrier headings

As suggested, we added numbers for paragraphs and subsections to be more comprehensive.

Reviewer 2 Report

The review by Lefèvre-Utile and colleagues reviews the current knowledge about the five functional levels of the epidermal barrier: the physical barrier, the chemical barrier, the microbiome barrier, het immune barrier and the nervous barrier. The topic is interesting for a broad readership and for the scientific community with an interest in dermatology.

Major comments:

My general feeling about the current status of the review is that it reads as a summing up of facts and not really representing a clear vision on the current state of the art of the different epidermal barriers. In most cases the authors also stay quite superficial in their statements: e.g. the statements on autophagy (see also comment below), explanation on involvement of TLRs, NLRs and CLRs in the immunological barrier (which receptors are most important according to the authors and on what grounds? Etc). What are the connections between the different barriers? Cross-talk between nervous barrier and microbes, etc. I would like to encourage the authors to revise thoroughly both the structure and the content of the review to better bring their vision on the current status of knowledge of the different epidermal barriers (and point out open questions, things that are still not clear, etc). Therefore, they need to further massage the current text.

Although the authors introduce the 5 different barriers they want to discuss in the review, this organization is more difficult to follow in the paper. Titles of sections jump from Physical Barrier to Keratinization, Junctions, Melanocytes, etc. It would be more comprehensive for the reader to keep the main titles referring to the different barriers discussed (subsections are possible according to the insight of the authors).

It would be interesting to make a better distinction between the outside-in barrier and the inside-out barrier of the epidermis (e.g. some mouse models have an intact outside-in barrier while having a leaky inside-out barrier).

Page 49: explain in one or two sentences the stratum lucidum.

Lines 78-86: If the authors want to discuss aspects of autophagy that prevent cell death, they should explain much better. Ref 11 in support of the statement that autophagy prevents premature cell death during terminal differentiation is more correct. They should explain better the experimental evidence for the statements made on the relation between autophagy and keratinocyte differentiation. In addition, also discuss conflicting data in literature (e.g keratinocyte-specific Beclin KO mice, Nogachi 2019 vs Wang 2020).

Line 97 and 101: ‘the mortar’ is in line 97 a lipid/protein matrix and in line 101 a lipid-rich matrix composed of ceramides, cholesterol and FFA. In the second definition proteins are not mentioned. Please explain better.

Line 234-241: no refs for the statements made in this part.

Figure 4. The message from this figure is far from clear, it is just mentioning the players on the field but not how they play together. Therefore, the added value of the figure in its current status is not clear and should be improved. Also, provide a decent legend.

Conclusions: Could the effect of the anti-inflammatory treatments on the expression of keratinocyte differentiation markers not just be due to the reduction of inflammatory cytokines in the skin under these conditions? For several inflammatory cytokines it has been shown they can suppress the expression of keratinocyte differentiation markers.

Editorial comments:

Title: The Five Functional Epidermal Barriers

Table 1: Cathepsin C

Table 1: the abbreviations mentioned below the table should be mentioned in the legend of the table.

Line 87: reference [Koenig 2020] should be numbered.

Author Response

Response to reviewer #2

The review by Lefèvre-Utile and colleagues reviews the current knowledge about the five functional levels of the epidermal barrier: the physical barrier, the chemical barrier, the microbiome barrier, the immune barrier and the nervous barrier. The topic is interesting for a broad readership and for the scientific community with an interest in dermatology.

General comment: My general feeling about the current status of the review is that it reads as a summing up of facts and not really representing a clear vision on the current state of the art of the different epidermal barriers. In most cases the authors also stay quite superficial in their statements: e.g. the statements on autophagy (see also comment below), explanation on involvement of TLRs, NLRs and CLRs in the immunological barrier (which receptors are most important according to the authors and on what grounds? Etc). What are the connections between the different barriers? Cross-talk between nervous barrier and microbes, etc. I would like to encourage the authors to revise thoroughly both the structure and the content of the review to better bring their vision on the current status of knowledge of the different epidermal barriers (and point out open questions, things that are still not clear, etc). Therefore, they need to further massage the current text.

We thank the reviewer for the careful reading of our manuscript. The goal was merely to present an overview of the different functional aspects of the epidermal barrier. As indicated, interactions between immune cells, sensory nerves and microbes are complex and involve protease-activated receptors (PARs) by endogenous and exogenous, microbiome-derived proteases, specific pattern recognition receptors (TLR, NLR, CLR) and the release of cytokines, neuropeptides, and neurotrophins by both keratinocytes, immune cells and neurons. In the present version, we comment on the known and potential interactions, stressing the need for further research.

Comment 1: Although the authors introduce the 5 different barriers they want to discuss in the review, this organization is more difficult to follow in the paper. Titles of sections jump from Physical Barrier to Keratinization, Junctions, Melanocytes, etc. It would be more comprehensive for the reader to keep the main titles referring to the different barriers discussed (subsections are possible according to the insight of the authors).

As suggested, we added numbers for paragraphs and subsections to be more comprehensive. We also modified the Title and the outlay, to better fit with the concept of various functional aspects of the epidermal barrier.

Comment 2: It would be interesting to make a better distinction between the outside-in barrier and the inside-out barrier of the epidermis (e.g. some mouse models have an intact outside-in barrier while having a leaky inside-out barrier).

We thank the reviewer for this pertinent comment. In the new version, focused on the functional aspects, we have introduced, where appropriate, variations related to the specific roles fulfilled by the epidermis.

Comment 3: Page 49: explain in one or two sentences the stratum lucidum.

As suggested, we added the following information: “The stratum lucidum is composed of 2 to 3 translucent layers of dead and anucleate keratinocytes and is only present in the thick and hairless skin of palms and soles”. Please refer to lines 63 to 65 in the revised manuscript.

Comment 4: Lines 78-86: If the authors want to discuss aspects of autophagy that prevent cell death, they should explain much better. Ref 11 in support of the statement that autophagy prevents premature cell death during terminal differentiation is more correct. They should explain better the experimental evidence for the statements made on the relation between autophagy and keratinocyte differentiation. In addition, also discuss conflicting data in literature (e.g keratinocyte-specific Beclin KO mice, Nogachi 2019 vs Wang 2020).

We agree that the present knowledge of exact intricate mechanisms leading to the controlled cornification is not yet complete. Consequently, we replaced the problematic paragraph by the following statement:

“Although evidence exists for a role of proteases, nucleases, and transglutaminases contributing to organelle disintegration, cell death and cornification, the molecular coordination of this mode of programmed cell death is far from being fully understood. Interestingly, it has been demonstrated that keratinocytes activate non-apoptotic and non-necroptotic pathway, i.e. autophagy, to prevent premature cell death during terminal differentiation. Indeed, disorganized apoptotic or necrotic cell death would lead to the release of DAMPs (danger-associated molecular patterns), inflammation and disturbance of the differentiation process of adjacent keratinocytes”. Please refer to lines 88 to 97 of the revised manuscript.

Comment 5: Line 97 and 101: ‘the mortar’ is in line 97 a lipid/protein matrix and in line 101 a lipid-rich matrix composed of ceramides, cholesterol, and FFA. In the second definition, proteins are not mentioned. Please explain better.

We thank the reviewer for his/her comment and agree that the paragrapher about the “mortar model” could be clarified. Accordingly, this paragrapher has been edited. We added the following information: In particular, we added that “The SC extracellular matrix contains not only lipids but also enzymes, structural proteins, and antimicrobial peptides [lines 103-104].”

Comment 6: Line 234-241: no refs for the statements made in this part. References were added.

Comment 7: Figure 4. The message from this figure is far from clear, it is just mentioning the players on the field but not how they play together. Therefore, the added value of the figure in its current status is not clear and should be improved. Also, provide a decent legend.

We thank the reviewer for to highlight these points. Indeed, to enhance the added value of Figure 4, we redesigned it with more details focusing on the interaction between the different aspects of the functional barrier. Figure legend has also been improved.

Comment 8: Conclusions: Could the effect of the anti-inflammatory treatments on the expression of keratinocyte differentiation markers not just be due to the reduction of inflammatory cytokines in the skin under these conditions? For several inflammatory cytokines it has been shown they can suppress the expression of keratinocyte differentiation markers.

We agree with this comment. The anti-inflammatory treatments are able to modulate the expression of keratinocyte differentiation markers through the inhibition of inflammatory cytokines. We did not develop further since there is already a review on this subject (Yang 2020) published in this journal. We keep the following information in our conclusion: « It has been recently demonstrated that the phosphodiesterase-4B inhibitor crisaborole, the  IL4 receptor antagonist dupilumab, and JAK-STAT inhibitors which are involved in the regulation of the immunological barrier are also able to increase the expression of filaggrin, loricrin, and claudins, thus contributing to the restoration of the physical barrier [82-84].»

Editorial comments:

Title: The Five Functional Epidermal Barriers: OK

Table 1: Cathepsin C: OK

Table 1: the abbreviations mentioned below the table should be mentioned in the legend of the table. OK

Line 87: reference [Koenig 2020] should be numbered. OK, #17.

Round 2

Reviewer 2 Report

The authors did several efforts to improve the paper in line with the comments raised by the referee.

However, there are still some items the authors should clarify or correct.

Figure 1: It would be good to add the basement membrane to anchor the hemidesmosomes in the basal layer in figure 1. I should have mentioned that in my initial comments.

Line 64: I guess it should read “….particular, colorless staining properties at….” (no comma between colorless and staining).

Line 67: by changing the sentence during revision ‘‘…with filaggrin (FLG)…’ can be removed.

Line 79: ….beneath…

Line 90: in the rebuttal letter another text is cited framing why the authors want to discuss autophagy. This text seems to be lacking in the revised version of the manuscript? I also think the authors should do somewhat more effort and put some nuance in the statement (the one mentioned in the rebuttal letter) that autophagy prevents the cells from premature death. How do the authors than explain the phenotype of ATG7 epidermal-specific KO mice (work of the L. Eckhart lab). Epidermal autophagy is absent in these mice, however there is no premature cell death observed in the epidermis of ATG7 deficient epidermis. Similar remark for ATG5 epidermis-specific ablation (Qiang et al., 2021).

The authors mention the conflicting data observed in two different models of keratin-specific beclin KO mice, but they forget to share their vision on these opposing phenotypes. Any way to reconcile? Possible explanations??

Line 98: “Protein and lipid envelopes 98 surround each brick,…”

Line 177: “…precocious…”

Line 259-260: “…., which can act as danger signals by activating keratinocytes.”

Line 268: “Role of the microbiome in skin”

Line 269: “…colonizes the epidermal surface.”

Line 361: also mention that ENaC is also involved (directly or indirectly) in lipid formation and secretion in the epidermis (work of the E. Hummler lab).

Author Response

Point-by-point responses

Manuscript Number: ijms-1348894
Title of Article: The Five Functional Epidermal Barriers
Name of the Corresponding Author: Pr. Francois Aubin
Email Address of the Corresponding Author: [email protected]

Specific responses

Response to reviewer #1

The authors did several efforts to improve the paper in line with the comments raised by the referee. However, there are still some items the authors should clarify or correct.

Figure 1: It would be good to add the basement membrane to anchor the hemidesmosomes in the basal layer in figure 1. I should have mentioned that in my initial comments. As suggested, we modified the Figure 1.

Line 64: I guess it should read “….particular, colorless staining properties at….” (no comma between colorless and staining).

This has been modified.

Line 67: by changing the sentence during revision ‘‘…with filaggrin (FLG)…’ can be removed.

This has been modified.

Line 79: ….beneath… This has been modified

Line 90: in the rebuttal letter another text is cited framing why the authors want to discuss autophagy. This text seems to be lacking in the revised version of the manuscript? I also think the authors should do somewhat more effort and put some nuance in the statement (the one mentioned in the rebuttal letter) that autophagy prevents the cells from premature death. How do the authors than explain the phenotype of ATG7 epidermal-specific KO mice (work of the L. Eckhart lab). Epidermal autophagy is absent in these mice, however there is no premature cell death observed in the epidermis of ATG7 deficient epidermis. Similar remark for ATG5 epidermis-specific ablation (Qiang et al., 2021).

The authors mention the conflicting data observed in two different models of keratin-specific beclin KO mice, but they forget to share their vision on these opposing phenotypes. Any way to reconcile? Possible explanations??

Based on the different comments, we decided to re-write this part. Because of conflicting data on the effect of autophagy in different models, we did not enter in details in our review. The text was modified as follows:

“Although evidence exist for a role of organelle disintegration, proteases, nucleases, and transglutaminases contributing to cell death, the molecular coordination of this mode of programmed cell death is far from understood [15-17]. Disorganized apoptotic or necrotic cell death would lead to the release of DAMPs (danger-associated molecular patterns including self antigens), inflammation autoimmunity and disturbance of the differentiation process of adjacent keratinocytes. Interestingly, it has been demonstrated that keratinocytes may activate non-apoptotic and non-necroptotic pathway, i.e. autophagy, to prevent premature cell death during terminal differentiation [15-17]. In addition, autophagy is involved in the immune response and neutralize pathogens and clear senescent cells. However, the autophagy machinery can either protect or cause autoimmune disorders. The physiological circumstances that govern this “balancing act” are not well understood. The cytokine microenvironment from dying immune cells, pathogens, or senescent cells can potentially direct the autophagic response in skin. It has been suggested that autophagy may participate in various physiological activities to ensure the smooth and quiescent operation of the immunotolerant environment and to maintain skin integrity [Sil, 2018]. »

Line 98: “Protein and lipid envelopes 98 surround each brick,…” This has been modified

Line 177: “…precocious…” This has been modified

Line 259-260: “…., which can act as danger signals by activating keratinocytes.” This has been modified

Line 268: “Role of the microbiome in skin” This has been modified

Line 269: “…colonizes the epidermal surface.” This has been modified

Line 361: also mention that ENaC is also involved (directly or indirectly) in lipid formation and secretion in the epidermis (work of the E. Hummler lab). This has been added (Charles et al, 2008).
